# Cross-Cultural Adaptation and Psychometric Properties of the Spanish Version of the OPQOL-Brief

**DOI:** 10.3390/ijerph20032062

**Published:** 2023-01-23

**Authors:** Natalia Perogil-Barragán, Santiago Gomez-Paniagua, Jorge Rojo-Ramos, María José González-Becerra, Sabina Barrios-Fernández, Konstantinos Gianikellis, Antonio Castillo-Paredes, Julián Carvajal-Gil, Laura Muñoz-Bermejo

**Affiliations:** 1University Hospital of the Canary Islands (HUC), 38001 Santa Cruz de Tenerife, Spain; 2BioẼrgon Research Group, University of Extremadura, 10003 Cáceres, Spain; 3Physical Activity for Education, Performance and Health (PAEPH) Research Group, Faculty of Sports Sciences, University of Extremadura, 10003 Cáceres, Spain; 4Social Impact and Innovation in Health (InHEALTH) Research Group, University Centre of Mérida, University of Extremadura, 06800 Mérida, Spain; 5Grupo AFySE, Investigación en Actividad Física y Salud Escolar, Escuela de Pedagogía en Educación Física, Facultad de Educación, Universidad de Las Américas, Santiago 8370040, Chile

**Keywords:** quality of life, OPQOL-Brief, transcultural adaptation, validity, older adults

## Abstract

Background: Ageing and its consequences on quality of life is one of the main issues to be addressed by public organizations; therefore the development of tools for its evaluation is a priority issue to orientate lines of action. Therefore, the objective of this research is to carry out cross-cultural adaptation and present the psychometric properties of the Spanish version of the Older People’s Quality of Life Questionnaire-Brief (OPQOL-Brief). Methods: a cross-cultural adaptation was carried out together with a process of translation and back-translation of the scale. The OPQOL-Brief and a sociodemographic questionnaire were administered to 120 older adults aged over 65 from a healthcare setting in the region of Extremadura. Results: the exploratory analysis revealed a factor structure through a matrix of polychoric correlations divided in two intercorrelated factors, consisting of 12 items and with excellent sample adequacy indices (KMO = 0.846’; Bartlett test = 1268.1; *p* = 0.000). Then, the confirmatory factor analysis established the definitive model with exceptional goodness-of-fit indices (NNFI = 0.99; CFI = 0.99; CMIN/DF = 0.96; Ρ (*χ*^2^) = 0.22; RMSEA = 0.037 and RMSR = 0.040). Finally, Cronbach’s alpha and McDonald’s omega for internal consistency reported good values of 0.830 and 0.851, respectively. Conclusion: our findings show that the OPQOL-Brief exhibits a solution with 12 elements and related constructs, providing stable goodness-of-fit indicators as well as good and remarkable consistency ratings.

## 1. Introduction

Population ageing and the rise of chronic diseases in developed countries have been widely accepted as major challenges for global public health [1]. Current demographic projections expect the proportion of elderly people will continue growing, and the very old segment will grow even faster. To meet the challenges of this situation, policies and services are increasingly focused on living in the community rather than relying on institutions as the main focus of care [2], improving the sustainability of care systems and the quality of life of users [3]. Thus, it is important to assess the cost effectiveness of interventions targeted for older people to identify those interventions with the strongest capacity to enhance older peoples’ quality of life and to provide value for money [4].

Ageing is a phenomenon present throughout the life cycle. However, despite being a natural process known to all, it is difficult to accept it as a reality innate to every human being [5]. In this sense, we could speak of normal or physiological ageing, which would be the expected changes in relation to the age of the individual, and of pathological ageing, which would be determined by the action of external agents that generate morbidity [6]. Nowadays, ageing means adapting to a society of accelerated and unexpected changes which means that, on many occasions, older people do not have the necessary resources to understand situations that involve new values and new ways of seeing and acting in the face of reality [7]. Traditionally, the age of 65 is considered to be the beginning of old age in developed countries because it is identified as the end of working life, although some authors consider that it should be raised, since the situation in which people reach this age today has nothing to do with earlier times [8]. For this reason, since the middle of the twentieth century, psychology and gerontology have been advocating for the need to create a new paradigm for analyzing and dealing with old age and the entire ageing process [9]

“Quality of life” is a major concept related with medical, social and psychological experimentation; also it is an important endpoint in the evaluation of public policies [10]. Nevertheless, this term is usually used as an umbrella concept, focusing its attention on a physical component, occasionally extended to a psychological component [11]. Thus, it is frequently not comprehended or determined, nor is there a consensus definition [12]. However, a series of aspects or factors that this term undoubtedly groups together can be established: (1) it is a multidimensional concept [13]; (2) it is dynamic, varying between and within individuals over time [14]; and it is composed of both objective and subjective components [15]. In 1995 the World Health Organization defined quality of life as “those perceptions by the individual of his or her position in life in the context of the culture and value system in which they live and in relation to their goals, expectations, standards and concerns” [16].

Successful ageing is a state where an individual is able to invoke psychological and social adaptive mechanisms to compensate for physiological limitations, to achieve a sense of well-being and a high self-assessment of quality of life [17].

Quality of life is considered one of the main objectives of healthcare, which once again demonstrates the importance of this concept and the need to study it. The assessment of quality of life is essential in the comprehensive assessment of the patient as it focuses more on the person than on the pathology itself, giving more importance to what the patient expresses than to the results of clinical tests. Moreover, by assessing it, we will be able to carry out health interventions and plan different resources that are necessary and useful. As life expectancy increases, valid measurement of the quality of life of people during these additional years is important as it has been shown to be a major predictor of mortality and other adverse health events [18]. When a concept cannot be measured directly, as is the case with quality of life, a series of questions are asked about different aspects of the concept, and a scale is created and tested for reliability, validity and sensitivity. With the growing interest in measuring quality of life in general, there is a need to create shorter measurement scales, with the desire to minimize the research burden, the number of questions and the number of respondents.

Despite of the large number of instruments developed and their application, evaluating and assessing quality of life, taking into account all the variables that influence it, is very complex [19]. At present, there are different scales and questionnaires for this purpose, among which we can highlight the World Health Organization Quality of Life–Old (WHOQOL-OLD) [20], the Older People´s Quality of Life Questionnaire (OPQOL-35) [21], the EuroQoL 5D-5L [22] or the Older People´s Quality of Life Questionnaire-Brief (OPQOL-Brief) [23]. OPQOL-Brief includes all domains of OPQOL-35 except religion/culture. It is a unique short scale that is less time consuming and reduces the research burden. In addition, OPQOL-Brief is an original measure that is developed from older people’s own thoughts. However, assessing quality of life is a difficult task as it encompasses multiple aspects, both objective and subjective, which can be interpreted in different ways. It is important to validate the OPQOL-Brief scale to see if there is a good relationship between its use in generic populations its use in specific populations such as older people. Currently, in Spain, different scales are used to measure quality of life, but none of these scales is specifically focused on the elderly. For this reason, there is a need for a validated instrument in Spanish to assess the quality of life of older people considering all the aspects that can influence it and the special characteristics of this group, in a simple, quick and comfortable way for both the interviewee and the interviewer.

Therefore, this study aims to carry out a translation and cross-cultural adaptation to the Spanish language, as well as present the OPQOL-Brief questionnaire factor structure and reliability to offer a validated assessment tool to evaluate the older adult´s quality of life in Spain, enabling public health organisations to establish lines of action and programmes oriented towards healthy ageing.

## 2. Materials and Methods

### 2.1. Study Design

A methodological cross-sectional study was conducted among people over 65 years of age living in the region of Extremadura (Spain). This cross-sectional study was conducted according to the recommendations of the Consensus-based Standards for the Selection of Health Measurement Instruments (COSMIN) [24] for the analysis of psychometric properties and in accordance with the stages of cross-cultural adaptation [25].

### 2.2. Procedure

For the translation and cross-cultural adaptation of the OPQOL-Brief questionnaire, the five stages of direct and back-translation were followed (Figure 1), according to the guidelines described by Beaton and his colleagues [25]:Two bilingual translators, both of whom were native Spanish speakers, independently performed two translations from English to Spanish of the original OPQOL-Brief. Both translators were briefed on the concept of the questionnaire, while only one of the translators was familiar with the terminology of the healthcare setting. Both translators prepared a written report of their translation.The translations were reviewed and synthesized into a single Spanish translation.Two translations from Spanish to English were carried out by two other bilingual translators with English as their mother tongue. Also, both translators were briefed on the concept of the questionnaire, while only one of the translators had healthcare experience. In addition, both translators produced a written report of their translation.The written reports documented all stages of the process and resulted in the final version of the questionnaire.The validity of the questionnaire was tested among 10 people over 65 years by face-to-face semi-structured interviews, whose objective was to test the relevance, comprehensibility, acceptability and feasibility of the questionnaire. In the last step, the final version of the questionnaire was obtained.

### 2.3. Participants

A total of 120 elderly people participated in the research. Participants were selected from municipalities in Extremadura by means of non-probabilistic convenience sampling. [26]. Inclusion criteria were (1) persons over 65 years of age (2) without previous pathologies (3) who provided a signed informed consent for the study. Persons older than 65 with physical and/or mental disabilities were excluded from the sample.

The total sample comprised 120 out of 128 elderly people (response rate 93.75%).

The sample calculation took into account the general considerations for studying the validity of scales and questionnaires, which suggest selecting 4 to 10 participants per item, with at least 100 participants to ensure the stability of the variance-covariance matrix.

### 2.4. Data Collection and Instruments

For data collection, a sociodemographic questionnaire was designed with 3 items: gender, age and marital status to characterise the sample. In addition, the Older People’s Quality of Life Questionnaire-Brief, which consists of a total of 13 items, was used as an instrument, with one of them not involved in the total score, and grouped into two dimensions: (1) “Psychosocial well-being”, defined as a higher-order concept that includes both emotional or psychological well-being, as well as social and collective well-being; and (2) “Life restrictions and limitations”, which refer to the difficulties an individual may have in carrying out activities or engaging in life situations in the actual context in which they live. A 5-point Likert-type scale is applied to the 13 items with a neutral response, being 1 “strongly disagree”, 2 “disagree”, 3 “neither agree or disagree”, 4 “agree” and 5 “strongly agree”. The total score is calculated by adding up the responses, with the range of assessment being possible between 13 (worst possible quality of life) and 65 (best possible quality of life). In terms of reliability, the instrument has a Cronbach’s alpha of 0.856, being > 0.70 for each of the two factors [23].

Taking into account that some older people might have problems with vision or writing, a researcher from the social and healthcare field was trained to conduct the interviews individually.

Data collection was carried out during the months of April and June 2022. Participants’ responses were stored directly in a spreadsheet for further statistical analysis.

### 2.5. Ethical Considerations

The study was conducted following the guidelines of the Declaration of Helsinki and was approved by the Bioethics and Biosafety Committee of the University of Extremadura (protocol code: 26/2022). Each participant voluntarily provided written informed consent.

### 2.6. Statistical Analysis

The exploratory analyses were carried out using the free statistical package FACTOR v.10.10.02 (Rovira I Virgili University: Tarragona, ESP) [27], taking into account the ordinal structure of the data gathered via a 5-choice Likert scale. As sampling adequacy metrics, the Kaiser–Meyer–Olkin (KMO) and Bartlett tests of sphericity were chosen [28]. A polychoric correlation matrix [29] was utilised to account for the nature of the data, and the optimum number of dimensions was determined by the best implementation of parallel analysis [30]. The robust unweighted least squares (RULS) approach with Promin rotation [31] was employed for factor extraction, assuming a correlation between components [32].

The CFA was then carried out using the software package AMOS v.26.0.0 (IBM Corporation, Wexford, PA, USA). Items having loadings of less than 0.60, cross-loadings of more than 0.40, and communalities of less than 0.30 were then eliminated from factor analysis [33]. The chi-squared probability setting as appropriate non-significant values (*p* > 0.05) [34]; the comparative fit index (CFI) and the non-normed fit index (NNFI) [35]; the root mean square error of approximation (RMSEA) [36]; and the root mean square of residuals (RMSR) [37] were used to assess the goodness-of-fit. In addition, Cronbach’s alpha coefficient and McDonald’s omega were also chosen as parameters for evaluating the questionnaire’s final structure.

The RULS approach with Promin rotation was used to reveal the components linked to the explained variance [36], as well as the consistency of the EAP scores [37].

Goodness-of-fit indices will be used to determine the fit between the data and the model after developing the AFC from the structure obtained in the EFA [38].

## 3. Results

The characteristics of participants are shown in Table 1.

Firstly, item 12 was discarded. This was done because this item did not influence the final scale score. Based on eigenvalues, the RULS approach with Promin rotation revealed two components linked to the explained variance as well as the consistency of EAP scores. The sampling adequacy indicators were used to assess the EFA’s feasibility, and the results were positive (KMO test = 0.846; and Bartlett test = 1268.1; df = 78; *p* = 0.000). The Normalized Direct Oblimin rotation approach was used because the amount of kurtosis (kurtosis = 232.777; *p* = 0.000) demanded non-parametric procedures. Table 2 shows the rotated loading matrix for twelve items and two factors established from the Normalized Direct Oblimin rotation method.

Following the EFA, a factor structure consisting of twelve elements was generated. The polychoric correlation matrix that defines the questionnaire’s design is shown in Table 3.

In order to understand the number of factors in the questionnaire, a parallel analysis was carried out and a two-factor model was obtained. The twelve items were grouped into two correlated factors: (1) Psychosocial well-being; and (2) Life restrictions/limitations. Table 4 shows the structure matrix of the questionnaire, in which four items are associated with the first factor and eight with the second; all items showed factor loadings higher than 0.3.

According to the correlation matrix between factors of the OPQOL-Brief (Psychosocial well-being and Life restrictions/limitations) there is a correlation of 0.313.

The CFA was used to build a definite model (Figure 2) after the structure of the questionnaire was specified.

The final questionnaire structure, shown in Figure 2, is made up of twelve items divided into two factors, and it includes the following values, from left to right: (1) correlation between factors; (2) standardized regression weights; (3) squared multiple correlations of each variable; and (4) correlations between exogenous variables.

The goodness-of-fit indices after developing the CFA from the structure obtained in the EFA are shown in Table 5. The values reported an exceptional matching between the data and the model. NNFI and CFI over 0.95 represent an immaculate fit to the model. Furthermore, a CMIN/DF index of less than 2 can be considered appropriate, and chi-squared probability with non-significant values is excellent. The RMSEA is within acceptable limits, and the RMSR could be considered exceptional since the values are less than 0.08.

Cronbach’s alpha and McDonald’s omega consistency indices for the dimensional structure are included in Table 6, as well as the explained variance of rotated factors.

The final Spanish version of the OPQOL-Brief questionnaire is shown in Appendix A (Table A1. Spanish version of the OPQOL-Brief questionnaire).

## 4. Discussion

The OPQOL-Brief is a unique contribution to the assessment area, especially given the demand for shorter, quick and free tests by researchers, clinicians and practitioners who can deliver numerous measures. It is unique in its social relevance, as it is built on a paradigm of measuring development based on people’s preferences. The complete and short versions of the OPQOL include aspects of life where older persons are more susceptible, but which are not covered by other scales with similar characteristics [38].

The main contribution of the present study is the investigation of the psychometric properties of the OPQOL-Brief questionnaire to assess the quality of life of older adults in Spain, providing the factor model and the reliability of the questionnaire in order to offer a validated assessment tool. The results showed a factor structure with optimal goodness-of-fit indicators consisting of two interrelated dimensions: psychosocial well-being (4 items) and life constraints/limitations (12 items).

The evaluation of the psychometric properties of the OPQOL-Brief has shown a high degree of reliability. Cronbach’s alpha values of 0.888 for the total score and 0.730 and 0.930 for each of the factors, respectively, indicate a high reliability of the questionnaire. These results are similar to the values of the original OPQOL-Brief, where the Cronbach’s alpha value was 0.856 [23]. The Turkish version, which has a Cronbach’s alpha value of 0.867 [39], the Persian version, which was 0.829 [40], and the Czech version, which was 0.921 [41], all show similar results to those obtained in this study. Likewise, the validation of the OPQOL-Brief questionnaire in the nursing home population gives a value of 0.83 [42], in line with the previous results. Also, the McDonald’s omega alternative reliability index likewise displays correct and great values [43].

The evaluation of construct validity provided a factorial solution of two correlated factors (0.313), as in the original validation [23] (0.313) and in the Turkish validation [39], but in contrast to the Persian validation [40] which established the existence of three factors. On the other hand, in both the Czech version and the version validated in the nursing home population, the construct validity obtained a unifactorial solution [41]. In addition, the analysis of the items indicated the desirability of eliminating item 12 (“I feel lucky compared to most people”) from the original 13-item proposal, as opposed to previous validations that considered all items in the questionnaire should be retained. Haugan et al. [42], on the other hand, found a better reliability of the scale by excluding five items and recommended using an eight-item scale for the older adult group.

There are various limitations to this study. The number of people in the sample is restricted. Because all the subjects were from the Extremadura region, sociocultural variables may have influenced the outcomes. More evidence of the OPQOL-Brief psychometric qualities will be needed in the future. Online questionnaires, on the other hand, provide the advantages of lowering expenses, removing the interviewer in relation to the respondents, extending the sample, and making data collecting and processing easier from the researcher’s perspective. Recruiting a larger sample from other regions of Spain as a future line of research would be interesting to gain more evidence on the OPQOL-Brief strengths. The ageing of the population has led to the search for practical and valid measurement tools both to promote healthy ageing and well-being and to evaluate the outcomes of interventions developed to achieve it. Shorter instruments have the benefits of reducing the burden for the persons under study and the costs of research. The OPQOL-Brief addresses the need for a shorter measure to assess quality of life in old age.

The OPQOL-Brief could be used at various healthcare levels and with a variety of health professionals to better understand the perspectives of older adults regarding quality of life. Understanding their relevance for public health agencies in developing policies and actions aimed at improving the quality of life of people in their later stages of life is relevant to assessing their impact on the well-being of older people.

## 5. Conclusions

The OPQOL-Brief is a short, highly reliable and valid measure to assess the quality of life of older people in Spain. Our results reveal a solution composed of twelve items and explained by two factors.

The contribution of the study is to propose a reliable, brief and validated measurement instrument, which offers simplicity, ease of reading and understanding for the elderly respondents and, on the other hand, facilitates the work of the interviewer and the progress of research. This questionnaire offers a reliable tool to assess quality of life and subsequently establish lines of action and programmes aimed at healthy ageing in the field of gerontology.

## Figures and Tables

**Figure 1 ijerph-20-02062-f001:**
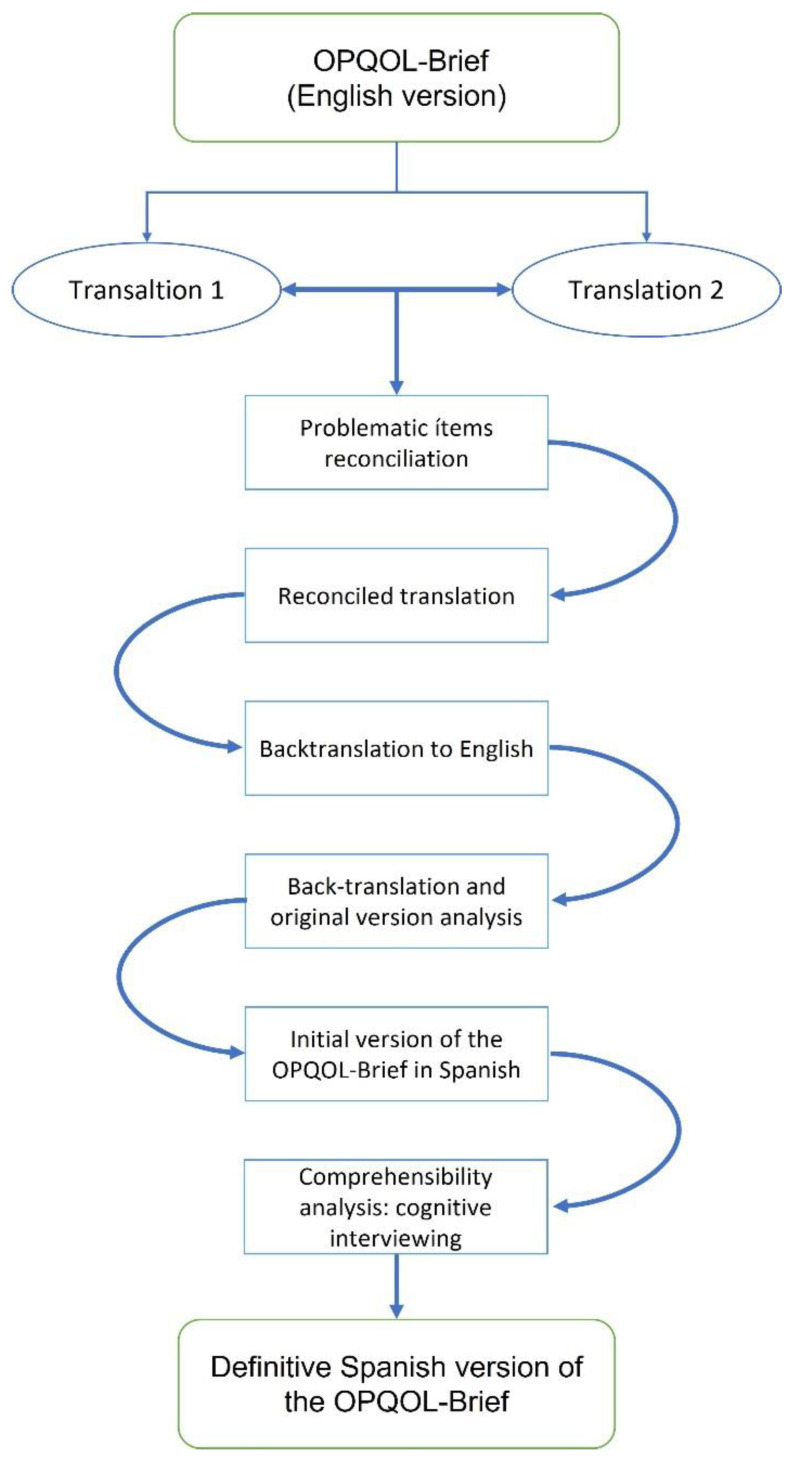
Translation and cultural adaptation process.

**Figure 2 ijerph-20-02062-f002:**
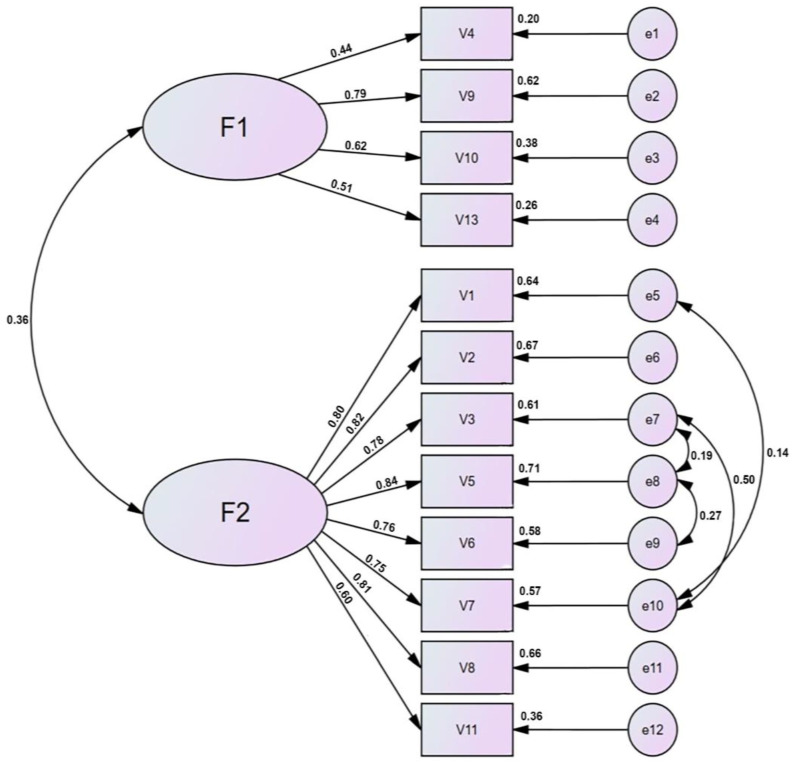
OPQOL-Brief factor structure.

**Table 1 ijerph-20-02062-t001:** Characteristics of the sample (*N* = 120).

Variables	Categories	*N*	%
Sex	Men	46	38.33
	Women	74	61.67
Age	Between 65 and 70	39	32.50
	Between 70 and 80	53	44.17
	Between 80 and 90	24	20.00
	Over 90	4	3.33
Marital status	Married	64	53.33
	Divorced	2	1.67
	Single	17	14.17
	Widowed	37	30.83
Marital status	Married	64	53.33
	Divorced	2	1.67
	Single	17	14.17
	Widowed	37	30.83

*N*: size of the subsample; %: percentage.

**Table 2 ijerph-20-02062-t002:** Rotated loading matrix.

Items	Factor 1	Factor 2
I enjoy my life overall.	0.259	0.725
2. I look forward to things.	−0.059	0.886
3. I am healthy enough to get out and about.	−0.014	0.873
4. My family, friends or neighbors will help me if needed.	0.472	0.093
5. I have social or leisure activities/hobbies that I enjoy doing.	0.042	0.862
6. I try to stay involved with things.	0.017	0.834
7. I am healthy enough to have my independence.	−0.217	0.927
8. I can please myself with what I do.	0.164	0.799
9. I feel safe where I live.	0.843	0.048
10. I get pleasure from my home.	0.807	−0.069
11. I take life as it comes and make the best of things.	0.336	0.517
12. I feel lucky compared to most people.	Deleted	
13. I have enough money to pay for household bills.	0.514	0.106

**Table 3 ijerph-20-02062-t003:** Polychoric correlation matrix for 12 items.

Items	1	2	3	4	5	6	7	8	9	10	11	13
1	1.000											
2	0.683	1.000										
3	0.650	0.736	1.000									
4	0.324	0.202	0.242	1.000								
5	0.777	0.798	0.677	0.213	1.000							
6	0.628	0.764	0.731	0.317	0.811	1.000						
7	0.597	0.705	0.871	0.057	0.681	0.699	1.000					
8	0.805	0.696	0.769	0.247	0.716	0.646	0.739	1.000				
9	0.407	0.192	0.283	0.476	0.334	0.288	0.096	0.375	1.000			
10	0.357	0.117	0.155	0.379	0.114	0.120	0.018	0.285	0.624	1.000		
11	0.614	0.568	0.485	0.153	0.560	0.524	0.491	0.526	0.466	0.482	1.000	
13	0.282	0.192	0.246	0.294	0.292	0.224	0.161	0.325	0.516	0.431	0.229	1.000

**Table 4 ijerph-20-02062-t004:** OPQOL-Brief rotated factor solution and factors loadings.

Items	Factor 1	Factor 2
I enjoy my life overall.	-	0.725
2. I look forward to things.	-	0.886
3. I am healthy enough to get out and about.	-	0.873
4. My family, friends or neighbors will help me if needed.	0.472	-
5. I have social or leisure activities/hobbies that I enjoy doing.	-	0.862
6. I try to stay involved with things.	-	0.834
7. I am healthy enough to have my independence.	-	0.927
8. I can please myself with what I do.	-	0.799
9. I feel safe where I live.	0.843	-
10. I get pleasure from my home.	0.807	-
11. I take life as it comes and make the best of things.	-	0.517
12. I feel lucky compared to most people.	Deleted	
13. I have enough money to pay for household bills	0.514	-

**Table 5 ijerph-20-02062-t005:** Goodness-of-fit indices.

Indices	Values
NNFI	0.99
CFI	0.99
CMIN/DF	0.96
Ρ (*χ*^2^)	0.22
RMSEA	0.037
RMSR	0.040

NNFI: non-normed fit index; CFI: comparative fit index; CMIN/DF: minimum discrepancy per degree of freedom; P (*χ*^2^): chi-squared probability; RMSEA: root mean square error of approximation; RMSR: root mean square of residuals.

**Table 6 ijerph-20-02062-t006:** Internal consistency of the OPQOL-Brief questionnaire.

Indexes	Psychosocial Well-Being	Life Restrictions/Limitations
Cronbach’s Alpha	0.730	0.930
McDonald’s Omega	0.764	0.938
Explained Variance	2.222	5.414

## Data Availability

The study was conducted according to the guidelines of the Declaration of Helsinki and approved by the Bioethics and Biosafety Committee at the University of Extremadura (protocol code: 26/2022).

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
