# Peer review of "Cross-Cultural Adaptation and Psychometric Properties of the Spanish Version of the OPQOL-Brief"

_ijerph, 2023, doi:10.3390/ijerph20032062_

Round 1

Reviewer 1 Report

The article concerns the adaptation of the OPQOL-Brief scale to the Spanish version. The authors may consider the following comments:

The article is generally well-executed formally, logically, linguistically, statistically and technically. Statistical validation indicates a translation work well done. in the language version:

Line 144. start a sentence with a capital letter.

Line 150: " finally the final sounds weird, I suggest changing it.

Figure1. title: missing T.

Line 182 can be added "discarded as it..." or divide it on two separate sentences.

Table 5. It's pointless to create a table to show only one variable, I suggest to insert the variable into the text.

In my opinion, the article could be accepted after the indicated changes.

Author Response

Dear Reviewer We appreciate your comments and recommendations to improve our manuscript. The detailed comments are as follows: REVIEWER 1 The article concerns the adaptation of the OPQOL-Brief scale to the Spanish version. The authors may consider the following comments: The article is generally well-executed formally, logically, linguistically, statistically and technically. Statistical validation indicates a translation work well done. in the language version: We appreciate your feedback. It encourages us to improve the manuscript as much as possible. Line 144. start a sentence with a capital letter. Thank you very much for your comment, it is a mistake that has been resolved. Line 150: " finally the final sounds weird, I suggest changing it. Thank you very much for your appreciation. The sentence has been modified: In the last step, the final version of the questionnaire was obtained. Figure1. title: missing T. Thank you for pointing out the error. Line 182 can be added "discarded as it..." or divide it on two separate sentences. Thank you for your comment. It has been split into two different sentences. Table 5. It's pointless to create a table to show only one variable, I suggest to insert the variable into the text. Thank you for your input. We have replaced table 5 with a comment referring to the correlation between the two factors of the OPQOL-Brief questionnaire.

Reviewer 2 Report

Comments

This study translates and culturally adapts the OPQOL-Brief to the Spanish elderly population and presents the factor structure and reliability. Although limited by the sample size and the region of the sample, it makes a unique contribution to public health by providing a QOL assessment instrument adapted to the Spanish elderly population.

1. Please explain why none of QOL scales was specifically focused on the elderly in Spain (ln95-96). Was there some difficulty or lack of need?

2. Please explain what had to be changed in the OPQOL-Brief, in addition to mere translation, to adapt it to the Spanish language and Spanish cultural characteristics. If there were any difficulties and efforts were made to overcome them, please indicate them in detail.

Author Response

Dear Reviewer We appreciate your comments and recommendations to improve our manuscript. The detailed comments are as follows: REVIEWER 2 This study translates and culturally adapts the OPQOL-Brief to the Spanish elderly population and presents the factor structure and reliability. Although limited by the sample size and the region of the sample, it makes a unique contribution to public health by providing a QOL assessment instrument adapted to the Spanish elderly population. 1. Please explain why none of QOL scales was specifically focused on the elderly in Spain (ln95-96). Was there some difficulty or lack of need? In older people, simple and short questionnaires are usually used to assess quality of life in order to carry out short evaluations over time and not to tire the respondents. One of the simplest and most widely used questionnaires is the EQ-5D-5L, validated in adults and used to assess quality of life also in older people in the Spanish National Health Survey. 2. Please explain what had to be changed in the OPQOL-Brief, in addition to mere translation, to adapt it to the Spanish language and Spanish cultural characteristics. If there were any difficulties and efforts were made to overcome them, please indicate them in detail. The OPQOL-Brief questionnaire was translated and adapted cross-culturally during five stages of direct and reverse translation according to the guidelines described by Beaton et al. However, the manuscript has detailed the need to train a health and social care interviewer to help resolve any doubts or lack of vision or writing that any older person may present.

Reviewer 3 Report

In my opinion the paper was not satisfactorily clear, informative and provide a valuable source document for anyone requiring a primer to know and understand this issue. Namely, numerous shortcomings in the section Introduction (ie the study aim), Materials and Methods, Results and Discussion make this paper not appropriate for publication in this form. Finally, the biggest problem in this study is Еthical issues. Some comments are listed by sections.     

Line 2-3: Add to the title that it was a hospital/clinical population of the elderly.  

Section Introduction:    

The Introduction of the study is well described. The knowledge gap is outlined well for the most part.  

- Lines 100-104: But, although the aim of this study is of interest, it is necessary for the authors to explain in detail why they chose to conduct this research in the population of elderly persons in hospitals and not in the general population.     

Section Materials and Methods:     

- Lines 105-112: Reconstruct the entire Materials and Methods section, so that the following subsections are described in detail: Study setting, Study Design, Study Population (explain reasons why clinical population selected, which the departments in this hospital have been included, which categories of patients are covered by this research - with one determined disease or any disease, whether the severity of the disease was taken into account, whether there were people who were clinicaly examined and marked as healthy, enter all data in Table 1), Study Sample (eligible participants, inclusion and exclusion criteria), Study Sample Calculation (cite appropriate references), Participation rate and Response rate, Data Collection, Ethical Considerations.     

In general, overlapping of the Materials and Methods section and the Results section is not allowed.   

Section Results:   

- Line 111:  Check and correct the title of the Table 1, is N = 605 correct?    

- Lines 182-184, 208: Cite references 36 and 37, as well as 38, in subsection Statistical Analysis, those references are not the results of this manuscript. In the Results section, authors should describe only their own results. Namely, the cited references numbers 36-37 refer to the methodology that is applied in similar researches and, if applied, it must be correctly described in the subsection Statistical Analysis.  

- In the Results section, clearly state the findings of the parallel analysis.  

- Check the order of citations and descriptions for Tables.       

- Line 224-227: On Table 7 and the accompanying text, you must indicate in which values ``Explained Variance'' was measured/determined.   

Section Discussion:  

Discussion section in general has no logical flow and is not comprehensive. The results of this study were compared with the results of only 2 studies (references 41 and 42). Several other versions of this questionnaire exist and are widely available in the literature, which were published as results of testing the psychometric characteristics of the observed questionnaire in several other languages in the population of elderly people in several countries. The differences in the findings of this study and the cited studies were not satisfactorily explained. The discussion must be completely reconstructed.    

In addition to all the above-mentioned remarks, the biggest problem in this manuscript is the following:

- Compare what is written in that paper on Lines 155-160 with what is written on Lines 250-260. What is correct?

- Compare what is written on Lines 160-161 (`Data collection was carried out during January and March 2021.`), Line 282 (`Institutional Review Board Statement: Not applicable.`) and Lines 285-287 (`Data Availability Statement: The study was conducted according to the guidelines of the Declaration of Helsinki and approved by the Bioethics and Biosafety Committee at the University of Extremadura (protocol code: 26/2022).`).

First of all, the aforementioned consent came a year after this study had already been carried out. Second, it was the time of the most severe waves of the COVID-19 pandemic, which was neither mentioned nor explained in the paper. Third, nowhere in the paper is it stated that informed written consent for participation in the study was obtained from the respondents in this study.      

Author Response

Dear Reviewer We appreciate your comments and recommendations to improve our manuscript. The detailed comments are as follows: REVIEWER 3 In my opinion the paper was not satisfactorily clear, informative and provide a valuable source document for anyone requiring a primer to know and understand this issue. Namely, numerous shortcomings in the section Introduction (ie the study aim), Materials and Methods, Results and Discussion make this paper not appropriate for publication in this form. Finally, the biggest problem in this study is Еthical issues. Some comments are listed by sections. Line 2-3: Add to the title that it was a hospital/clinical population of the elderly. We appreciate your suggestion, but the participants were not recruited from hospitals and health centres. It was healthy people, without pathologies and over 65 years of age who answered the questionnaire. The reference to hospital population has been removed. Section Introduction: The Introduction of the study is well described. The knowledge gap is outlined well for the most part. - Lines 100-104: But, although the aim of this study is of interest, it is necessary for the authors to explain in detail why they chose to conduct this research in the population of elderly persons in hospitals and not in the general population. We agree that the wording in these lines may be misleading. In no case has the study been carried out on people admitted to hospitals or clinics, so the inclusion criteria have been detailed in the "Participants" section. Section Materials and Methods: - Lines 105-112: Reconstruct the entire Materials and Methods section, so that the following subsections are described in detail: Study setting, Study Design, Study Population (explain reasons why clinical population selected, which the departments in this hospital have been included, which categories of patients are covered by this research - with one determined disease or any disease, whether the severity of the disease was taken into account, whether there were people who were clinicaly examined and marked as healthy, enter all data in Table 1), Study Sample (eligible participants, inclusion and exclusion criteria), Study Sample Calculation (cite appropriate references), Participation rate and Response rate, Data Collection, Ethical Considerations. Thank you for this contribution. The new redistribution of the materials and methods section helps to improve the manuscript. In general, overlapping of the Materials and Methods section and the Results section is not allowed.
Section Results: - Line 111: Check and correct the title of the Table 1, is N = 605 correct? Thank you for your comment. This is an error. It has been changed (N=120). - Lines 182-184, 208: Cite references 36 and 37, as well as 38, in subsection Statistical Analysis, those references are not the results of this manuscript. In the Results section, authors should describe only their own results. Namely, the cited references numbers 36-37 refer to the methodology that is applied in similar researches and, if applied, it must be correctly described in the subsection Statistical Analysis. We agree to correctly apply the bibliographic references in the corresponding section "Statistical analysis". - In the Results section, clearly state the findings of the parallel analysis. The result of the parallel analysis referred to in table 4 has been included. - Check the order of citations and descriptions for Tables. Thank you for your comments. We have made some changes to the citations and tables so that they are in strict order in the text. - Line 224-227: On Table 7 and the accompanying text, you must indicate in which values ``Explained Variance'' was measured/determined. Explained variance is computed by squaring the items’ loadings of structure matrix and dividing it by residuals. As the questionnaire scores are based on a Likert scale, they do not have a specific unit. Section Discussion: Discussion section in general has no logical flow and is not comprehensive. The results of this study were compared with the results of only 2 studies (references 41 and 42). Several other versions of this questionnaire exist and are widely available in the literature, which were published as results of testing the psychometric characteristics of the observed questionnaire in several other languages in the population of elderly people in several countries. The differences in the findings of this study and the cited studies were not satisfactorily explained. The discussion must be completely reconstructed. We thank you for your feedback. The discussion section has been modified and additional background studies have been included. In addition to all the above-mentioned remarks, the biggest problem in this manuscript is the following: - Compare what is written in that paper on Lines 155-160 with what is written on Lines 250-260. What is correct? We understand your comment, the interviews were conducted in person by a face-to-face interviewer, therefore the limitation for line 250 has been removed.
- Compare what is written on Lines 160-161 (`Data collection was carried out during January and March 2021.`), Line 282 (`Institutional Review Board Statement: Not applicable.`) and Lines 285-287 (`Data Availability Statement: The study was conducted according to the guidelines of the Declaration of Helsinki and approved by the Bioethics and Biosafety Committee at the University of Extremadura (protocol code: 26/2022).`). First of all, the aforementioned consent came a year after this study had already been carried out. Second, it was the time of the most severe waves of the COVID-19 pandemic, which was neither mentioned nor explained in the paper. Third, nowhere in the paper is it stated that informed written consent for participation in the study was obtained from the respondents in this study.
We understand your confusion about the dates. This is a transcription error, as we conducted a preliminary survey on these dates. Data collection was carried out between April and June 2022. It has been amended in the manuscript. Informed consent was sought from each participant, as detailed in the paper

Round 2

Reviewer 3 Report

Due to very serious flaws and serious issues as it pertains to the study's methodology, study population, sample, ethical considerations, it is impossible to determine the credibility of the information, methodology and data within this manuscript. Namely, after the initial review and questions raised regarding inconsistencies in the paper, the authors have extremely changed around the information provided in the paper throughout the revision rounds. The study population and sample changed, the data collection changed, ethical issues remained questionable - first there were conflicting information provided regarding the existence or necessity of an ethical approval for conducting the study, differing information regarding participants' consent etc.    

Author Response

REVIEWER 3

The article concerns the adaptation of the OPQOL-Brief scale to the Spanish version. The authors may consider the following comments:

Due to very serious flaws and serious issues as it pertains to the study's methodology, study population, sample, ethical considerations, it is impossible to determine the credibility of the information, methodology and data within this manuscript. Namely, after the initial review and questions raised regarding inconsistencies in the paper, the authors have extremely changed around the information provided in the paper throughout the revision rounds. The study population and sample changed, the data collection changed, ethical issues remained questionable - first there were conflicting information provided regarding the existence or necessity of an ethical approval for conducting the study, differing information regarding participants' consent etc.   

We understand your concern about the study methodology. In the following, we justify some of the changes made, but in no case are they outside the veracity of the work carried out. The reference to the hospital population was eliminated, as the surveys were carried out in the vicinity of health centres and hospitals in the municipalities, but proximity to these institutions is not relevant to the methodology. The questionnaires were administered to healthy people, without pathologies and over 65 years of age. Inclusion criteria were not indicated in the first manuscript and were incorporated during the revision at the request of the reviewer.

The recommended sections were included in the methodology section, but the study sample has not changed in any way, but a misinterpretation of the scope of data collection has been corrected. In the first manuscript reference was made to ethical approval and its code and the date of data collection, which was obviously wrong, was corrected.

As mentioned above, we understand your concern, but all the data described in the study methodology section are accurate and we believe that the modifications made have improved the quality of the manuscript, as errors have been corrected and improvements have been incorporated into each section.
